# YOUR MIXTURE-OF-EXPERTS LLM IS SECRETLY AN EMBEDDING MODEL FOR FREE

**Ziyue Li, Tianyi Zhou**
Department of Computer Science
University of Maryland, College Park
{litzy619,tianyi}@umd.edu
Project: https://github.com/tianyi-lab/MoE-Embedding

## ABSTRACT

While large language models (LLMs) excel on generation tasks, their decoder-only architecture often limits their potential as embedding models if no further representation finetuning is applied. Does this contradict their claim of generalists? To answer the question, we take a closer look at Mixture-of-Experts (MoE) LLMs. Our study shows that the expert routers in MoE LLMs can serve as an off-the-shelf embedding model with promising performance on a diverse class of embedding-focused tasks, without requiring any finetuning. Moreover, our extensive analysis shows that the MoE routing weights (RW) is complementary to the hidden state (HS) of LLMs, a widely-used embedding. Compared to HS, we find that RW is more robust to the choice of prompts and focuses on high-level semantics. Motivated by the analysis, we propose MoEE combining RW and HS, which achieves better performance than using either separately. Our exploration of their combination and prompting strategy shed several novel insights, e.g., a weighted sum of RW and HS similarities outperforms the similarity on their concatenation. Our experiments are conducted on 6 embedding tasks with 20 datasets from the Massive Text Embedding Benchmark (MTEB). The results demonstrate the significant improvement brought by MoEE to LLM-based embedding without further finetuning.

## 1 INTRODUCTION

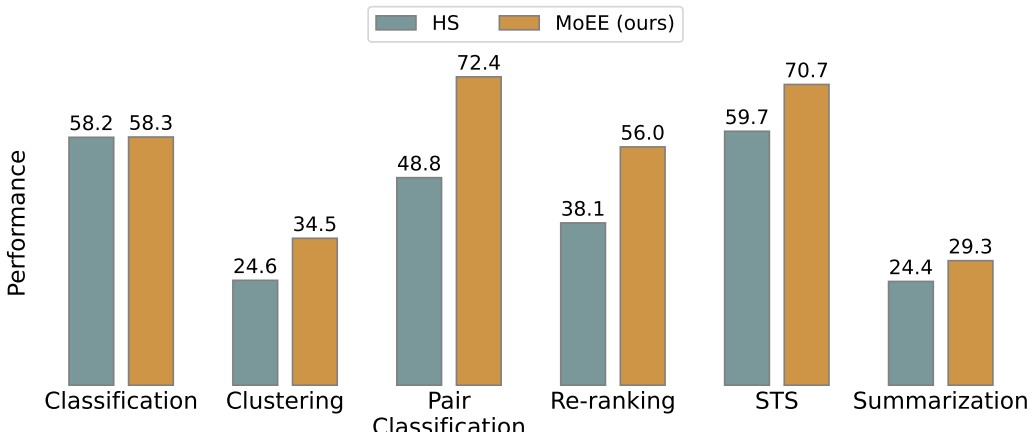

Figure 1: Comparison of hidden state (HS) and MoEE (ours) on six types of tasks from the Massive Text Embedding Benchmark (MTEB), where MoEE consistently outperforms HS on all tasks. Both HS and MoEE are extracted from DeepSeekMoE-16B (Dai et al., 2024) without further finetuning.

Mixture-of-Experts (MoE) (Jacobs et al., 1991; Jordan & Jacobs, 1994), as a versatile architecture originally developed in the 1990s, can improve model generalization and reduce inference cost by distributing tasks to specialized experts (Shazeer et al., 2017). Over time, MoE is gaining prominence in fields such as natural language processing (Shen et al., 2023) and computer vision (Li et al., 2023;

Zong et al., 2024; Lin et al., 2024; Shi et al., 2024), especially attracting growing attention in the development of large language models (LLMs) (Muennighoff et al., 2024a; Dai et al., 2024; Jiang et al., 2024). A key component of MoE is the dynamic routers, which intelligently assign each input to the most relevant expert. This allows MoE to tailor its computations to the unique characteristics of each input, optimizing both efficiency and accuracy.

However, most recent LLMs and MoE LLMs are built upon the decoder-only architecture trained for autoregressive next-token prediction. While excelling on generative tasks, their final or intermediate hidden state (HS) is not designed to capture the key features of input tokens and cover all their information. Instead, HS can be biased towards the information of the next output token. Although it is a common empirical practice to extract the last token's hidden state (HS) as embedding (Wang et al., 2024), it may even perform much poorer than smaller encoder models specifically trained for embedding tasks (Lei et al., 2024; Muennighoff et al., 2024b). Take classification as an example, inputs with subtly different semantics may be associated with the same label, so the last HS aiming to predict the label may ignore the input difference. Although extra finetuning specifically for representation learning (Lee et al., 2024; Muennighoff et al., 2024b) can greatly strengthen LLM's capability as an embedding model, it raises the question of whether pre-trained LLMs can be claimed as generalists, given the broad application of embedding tasks.

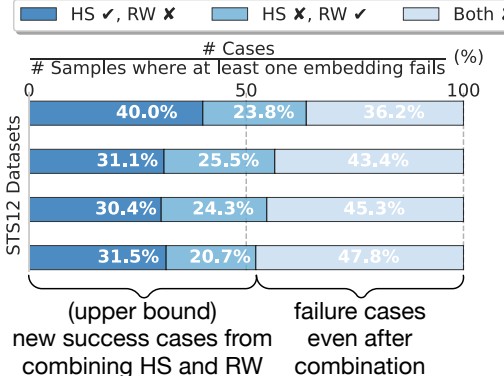

Figure 2: **Complementarity of DeepSeekMoE-16B's routing weights (RW) and hidden state (HS)** as embedding in the task of similarity ranking on STS12 datasets. In the error analysis of instances where at least one embedding fails[1], we report the proportion of (1) HS succeeds ✓ and RW fails ✗; (2) HS fails and RW succeeds, and (3) both RW and HS fail. In most cases, the proportion of (1)+(2) exceeds (3), indicating the complementarity of RW and HS.

*Can we extract high-quality embedding directly from LLMs without additional training?* In this paper, we find a Yes-answer to the question when studying MoE LLMs. Our main discovery is that **the routers in MoE can serve as an off-the-shelf embedding model and the produced routing weights (RW) provide complementary information to the widely used HS as embedding.** Compared to HS focusing on the final prediction results from the input, RW reflects the intermediate reasoning choices of MoE on the input for each layer of LLMs. Hence, as a byproduct of the routing mechanism, RW completes the input information missing in HS. As evidence, our comparative analysis of RW and HS shows that they reveal different clustering structures and topics of inputs, while RW captures the input's underlying themes and semantic structures. Moreover, we conducted an error analysis of the embedding task instances on which either HS or RW failed. As shown in Fig. 2, the proportion of cases where one embedding succeeds and the other fails exceeds $50\%$, indicating a large room for improvement if combining RW and HS.

Motivated by the analysis, we propose the first attempt to combine RW and the widely-used HS of MoE LLMs, resulting in a *training-free, contextual-rich, and holistic embedding* called "**MoE Embedding (MOEE)**" that excels in embedding tasks. Specifically, we experiment with various combination strategies and find that while simple concatenation of RW and HS (denoted by MOEE (concat)) improves either of them, a weighted sum of the two similarities computed on RW and HS separately (denoted by MOEE (sum)) often achieves the best results. The weighted sum of similarities avoids the fusion and alignment between the two different types of embedding while allowing us to balance output-dependent information with input-sensitive features, optimizing performance across diverse tasks.

We conduct extension evaluations of MOEE and compare it with baselines on the Massive Text Embedding Benchmark (MTEB) (Muennighoff et al., 2022), which covers a wide range of tasks designed to test embedding quality. MOEE consistently outperforms embedding derived solely from HS or MoE's

---

[1]Success/Failure is determined by how closely the ranking based on the embedding matches the ground truth, with deviations beyond a threshold marked as failures.

RW, as shown in Figure 1. Notably, MOEE (sum) achieves significant gains in tasks requiring an in-depth understanding of the input, such as semantic textual similarity, classification, and clustering.

The rest of the paper is organized as follows: §2 reviews related work on existing embedding methods and MoE. §3 outlines our methodology for integrating RW of MoE with the widely-used HS embedding. §4 reports experimental results on MTEB, highlighting MOEE's advantages on performance and interpretability. Finally, §5 discusses the implications and future research directions. Results in the paper except §4 are conducted on DeepSeekMoE-16B (Dai et al., 2024) unless specified.

## 2 RELATED WORK

**Training-Based Embedding (pre-LLM)** Early work on sentence embedding, such as SkipThought (Kiros et al., 2015), leveraged the distributional hypothesis by predicting surrounding sentences from a given input. These methods typically employed sequence-to-sequence architectures, following the success of Word2Vec (Mikolov, 2013). Recent advancements have shifted toward contrastive learning, which has gained prominence for its effectiveness in self-supervised representation learning. Contrastive methods, such as SimCSE (Gao et al., 2021), exploit different views of the same sentence through data augmentation or dropout, treating different outputs as positive pairs and negative pairs as unrelated sentences. This approach helps models better capture semantic similarities by maximizing the similarity between positive pairs while minimizing it between negative ones. Contrastive learning has been widely applied in sentence embedding due to its simplicity and competitive performance (Wu et al., 2020; Wang et al., 2021; Meng et al., 2021). Other methods like InfoNCE (Oord et al., 2018) and MoCo (He et al., 2020) have also contributed to the development of contrastive frameworks, further enhancing embedding quality. While effective, these approaches rely on static architectures that may overlook input variability. In contrast, MoE models dynamically route inputs through specialized experts, producing more nuanced, context-aware embedding without additional training.

**Training-Based Embedding with LLMs** Recent advances in language modeling have demonstrated the potential of LLMs to generate high-quality sentence embedding (Muennighoff et al., 2024b; Meng et al., 2024). For instance, some methods, such as Sentence-T5 (Ni et al., 2021), employ contrastive learning and are capable of generating embedding that rivals fine-tuned models, even with billions of parameters. However, these methods often depend on complex pretraining and large-scale contrastive objectives, limiting their flexibility for new tasks without retraining.

**Training-Free Embedding with LLMs** Training-free approaches seek to directly extract embedding from pre-trained LLMs without the need for additional finetuning. While this process is relatively straightforward for encoder-decoder models (Ni et al., 2021), it presents challenges for the more common decoder-only LLMs, where deriving meaningful embedding is less intuitive. Current approaches typically utilize the generated hidden state(s) of these models (Jiang et al., 2023). To improve the quality of these embedding, prompt-based techniques have gained traction (Jiang et al., 2022; Lei et al., 2024). One such method, Prompt with Explicit One Word Limitation (PromptEOL) (Jiang et al., 2023), distills sentence meaning into a compact embedding by prompting the model with the instruction: *'This sentence: "[text]" means in one word: '*.

In pre-trained decoder-only LLMs, embedding is typically derived from the hidden state of the final layer. Given an input sequence $\mathbf{x} = [x_1, x_2, \ldots, x_T]$, let $\mathbf{H}^{(l)} \in \mathbb{R}^{T \times d}$ represent the hidden state at the $l$-th layer, where $T$ is the sequence length, $d$ is the hidden state dimension, and $l = 1, 2, \ldots, L$ is the layer index.

To extract a single embedding $\mathbf{e}_{\text{HS}}$ that represents the entire input sequence, one approach is to use the last token's hidden state in the final layer, expressed as:

$$\mathbf{e}_{\text{HS}} = \mathbf{H}_T^{(L)} \in \mathbb{R}^d$$

Another approach is to apply pooling over all tokens in the last layer. For example, mean pooling averages the hidden states as: $\frac{1}{T} \sum_{i=1}^{T} \mathbf{H}_i^{(L)}$. These methods provide flexibility based on task requirements, with the resulting embedding capturing the context of the input sequence as modeled by the LLM.

**Mixture-of-Experts (MoE)** MoE models have been predominantly used in multitask learning and efficient large-scale training scenarios (Shazeer et al., 2017). However, their potential for generating

instance-level embedding has been underexplored. Our method leverages the routing decisions made by MoE models to generate embedding that are sensitive to the input's structure and semantics. This results in more flexible and interpretable embedding compared to static models, without the overhead of task-specific retraining.

# 3 Mixture-of-Experts Embedding (MoEE)

Our approach leverages the dynamic routing mechanisms of pre-trained, decoder-only LLMs equipped with MoE modules to generate enriched, input-sensitive embedding. This section details the key steps of our methodology, including embedding extraction, expert routing across layers, and the final integration of embedding—all achieved using pre-trained models without any additional training.

## 3.1 MoE Routing Weights (RW) as Embedding

Our approach capitalizes on the dynamic routing capabilities of MoE models embedded in pre-trained, decoder-only LLMs. These MoE modules operate across multiple layers, enabling the model to specialize in processing different aspects of the input at varying depths.

Each MoE model at layer $l$ consists of $N^{(l)}$ experts, denoted by $E_i^{(l)}$, where $i = 1, 2, \ldots, N^{(l)}$. Each expert is a specialized sub-network that focuses on specific input characteristics at that layer, allowing for a more granular understanding of the input as it passes through the network. However, the true strength of this architecture lies in the dynamic routing mechanism, governed by a gating function $\mathbf{g}^{(l)}(\mathbf{H}^{(l)}) \in \mathbb{R}^{N^{(l)}}$, which determines which experts will be activated at each layer based on the input.

This gating function outputs a probability distribution over the available experts in each layer, dynamically selecting the most relevant ones for the current input. The routing weights $g_i^{(l)}(\mathbf{H}^{(l)})$ indicate the contribution of each expert to the final output of layer $l$, formulated as: $\sum_{i=1}^{N^{(l)}} g_i^{(l)}(\mathbf{H}^{(l)}) E_i^{(l)}(\mathbf{H}^{(l)})$, where $\sum_{i=1}^{N^{(l)}} g_i^{(l)}(\mathbf{H}^{(l)}) = 1$, ensuring a weighted combination of experts. The gating function is typically implemented as a softmax over a set of logits $\mathbf{z}^{(l)}(\mathbf{H}^{(l)})$, making the routing decision both flexible and data-driven:

$$g_i^{(l)}(\mathbf{H}^{(l)}) = \frac{\exp(z_i^{(l)}(\mathbf{H}^{(l)}))}{\sum_{j=1}^{N^{(l)}} \exp(z_j^{(l)}(\mathbf{H}^{(l)}))}.$$

By leveraging the routing weights from **all layers**, our approach captures a richer representation of the input that accounts for both shallow and deep contextual features. This enables the model to provide nuanced information at every level of abstraction, which is critical for tasks requiring sensitivity to both low-level and high-level input details.

By concatenating the dynamic routing weights from all layers, we form a comprehensive routing-based embedding $\mathbf{e}_{\text{RW}}$:

$$\mathbf{e}_{\text{RW}} = [\mathbf{g}^{(1)}(\mathbf{H}^{(1)}); \mathbf{g}^{(2)}(\mathbf{H}^{(2)}); \ldots; \mathbf{g}^{(L)}(\mathbf{H}^{(L)})] \in \mathbb{R}^{\sum_{l=1}^{L} N^{(l)}}.$$

This embedding captures how the input is routed through different experts across all layers, offering a holistic view of the model's interaction with the input. Importantly, it reflects the full depth of the model's decision-making process, making it a powerful representation for downstream tasks where diverse semantic and structural features of the input are essential.

## 3.2 Comparative & Complementary Analysis of Routing Weights & Hidden State

In this section, we investigate how routing weight (RW) embedding and hidden state (HS) embedding, generated from MoE models, capture different aspects of input data. Understanding the distinct roles these embedding play is crucial to determining how they complement each other. While HS embedding from pre-trained LLMs provides a broad, context-driven representation of sentences, they may overlook the nuanced, token-specific information that RW embedding can capture through MoE's dynamic routing.

This distinction suggests that RW and HS may excel in different contexts, potentially encoding complementary information. To explore this, we first analyze their clustering behavior using k-means

Table 1: Correlation of the clustering results achieved on the routing weight (RW) and hidden state (HS) embedding extracted from MoE LLMs. Low scores indicate the complementarity of RW and HS.

| Metric | Score (*max value*) |
|---|---|
| Adjusted Mutual Information (AMI) | 0.29 (*1.00*) |
| Normalized Mutual Information (NMI) | 0.29 (*1.00*) |
| Jaccard Similarity | 0.06 (*1.00*) |
| Exact Matching (%) | 45.54% (*100.00%*) |

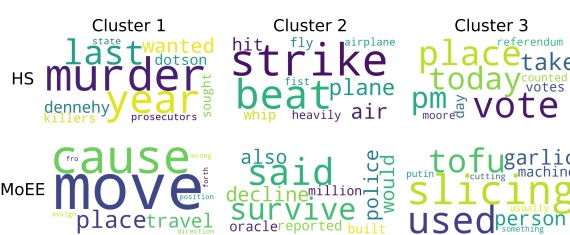

Figure 3: Word clouds of the top-20 topics from 3 clusters achieved on RW and HS separately, highlighting their captured distinct semantic features.

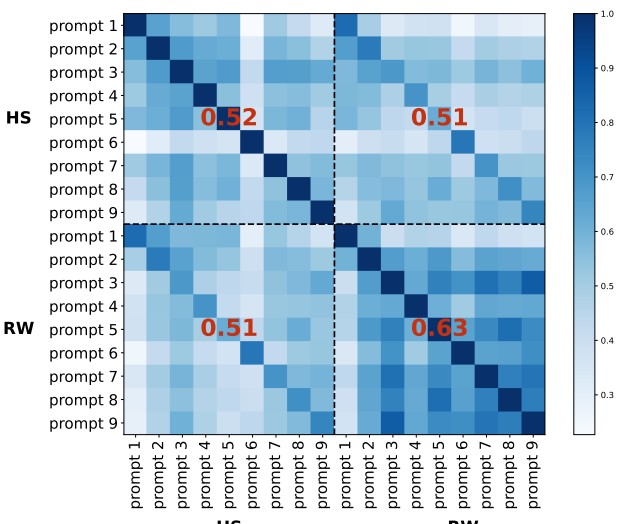

Figure 4: Heatmap of Spearman's rank correlation between RW and HS embedding achieved using nine different prompts (defined in Table 2). The top-left (HS-HS) and bottom-right (RW-RW) blocks show the correlations between embedding when using different prompts, with mean scores of 0.52 and 0.63 (excluding the diagonal entries), respectively. This implies **RW is more robust to varying prompts than HS**. The top-right and bottom-left blocks reflect correlations between RW and HS when using the same or different prompts, both with a mean score of 0.51. **This lowest score indicates the complementarity between RW and HS.**

Table 2: Prompts used in Fig 4-5.

| ID | Prompt |
|---|---|
| 1 | This sentence: *sent* means in one word: |
| 2 | In one word, describe the style of the following sentence - *sent*: |
| 3 | In one word, describe the sentiment of the following sentence (positive, neutral, or negative) - *sent*: |
| 4 | In one word, describe the tone of the following sentence - *sent* (e.g., formal, informal, humorous, serious): |
| 5 | In one word, describe the intent behind the following sentence (e.g., request, suggestion, command) - *sent*: |
| 6 | In one word, rate the complexity of the following sentence (simple, moderate, complex) - *sent*: |
| 7 | In one word, describe whether the following sentence is subjective or objective - *sent*: |
| 8 | In one word, describe the language style of the following sentence (e.g., academic, conversational, literary) - *sent*: |
| 9 | In one word, describe the grammatical structure of the following sentence (simple, compound, complex) - *sent*: |

clustering and perform a correlation analysis to quantify the differences between their respective cluster structures. We then leverage the BERTopic framework (Grootendorst, 2022) to examine the topics associated with each cluster, providing insights into the embedding's capacity to capture thematic content. Finally, we evaluate their performance in identifying semantically similar text pairs, further confirming their complementary nature.

**RW and HS embedding exhibit distinct clustering behaviors and encode different topics.** Our analysis shows that the clustering results from RW and HS embedding are markedly different. As reflected in Table 1, the clustering metrics show moderate overlap (AMI and NMI at 0.29), but with a low Jaccard Similarity of 0.06 and only 45.54% exact matching[2] between clusters, underscoring the distinct ways each method structures the data. This difference in clustering behavior is further reflected in the topics captured by the embedding. As shown in Figure 3, the word clouds reveal that

---

[2]Exact matching refers to the proportion of data points that are grouped into identical clusters by two different methods (in this case, RW and HS embeddings).

the clusters from RW and HS embedding emphasize different thematic topics, highlighting how the two methods capture divergent aspects of the input data.

**Complementary nature of RW and HS embedding.** Previous analyses suggest that RW and HS embedding capture different aspects of input data. To validate this hypothesis and quantify their complementarity, we need to examine how these two embedding relate to one another. We approach this by conducting a Spearman correlation analysis using the STS12 dataset, which contains 6,216 sentence pairs. For each pair, we generate embedding from both RW and HS and calculate the similarity between the sentences to assess how each embedding captures semantic relationships. To ensure that any observed differences are not caused by prompt variation, we employ nine distinct prompts (listed in Table 2). As shown in Figure 4, notably, the correlation between RW and HS embedding is the lowest across all comparisons, with a mean value of 0.51. This low correlation highlights that RW and HS capture largely unrelated aspects of the data, reinforcing their complementary nature. Further evidence supporting this complementarity is presented in the error analysis (Figure 2) and the experimental results (Section 4).

### 3.3 THE PROPOSED **MoE** EMBEDDING (MoEE)

Building on the analysis of routing weight (RW) and hidden state (HS) embedding, we propose our method MoEE, which combines RW and HS to form a more comprehensive embedding representation. We introduce two approaches for this combination as follows.

**Concatenation-based Combination.** In this method, the embedding generated by the hidden state ($\mathbf{e}_{\text{HS}}$) and the routing weights ($\mathbf{e}_{\text{RW}}$) are concatenated to form the final embedding. This approach is denoted as MoEE (concat), and the final embedding is computed as:

$$\mathbf{e}_{\text{final}} = [\mathbf{e}_{\text{HS}}; \mathbf{e}_{\text{RW}}] \in \mathbb{R}^{d_{\text{HS}}+d_{\text{RW}}},$$

where $d_{\text{HS}}$ is the dimensionality of the hidden state embedding, and $d_{\text{RW}}$ is the dimensionality of the routing weight embedding. This method preserves the distinct information captured by each component while allowing downstream tasks to leverage the combined representation.

**Weighted Sum Integration.** The second method performs a weighted sum of the similarity scores calculated from RW and HS embedding, denoted as MoEE (sum). For tasks like STS, given a sentence pair $(s_1, s_2)$, we first compute the similarity score between the two sentences using both HS-based embedding and RW-based embedding independently, as $\mathbf{e}_{\text{HS}}(s_1)$, $\mathbf{e}_{\text{HS}}(s_2)$, $\mathbf{e}_{\text{RW}}(s_1)$, and $\mathbf{e}_{\text{RW}}(s_2)$. Then, a weighted sum of the similarity scores is performed before comparing the result to the ground truth:

$$\text{sim}_{\text{HS}} = \text{cosine\_similarity}(\mathbf{e}_{\text{HS}}(s_1), \mathbf{e}_{\text{HS}}(s_2)),$$
$$\text{sim}_{\text{RW}} = \text{cosine\_similarity}(\mathbf{e}_{\text{RW}}(s_1), \mathbf{e}_{\text{RW}}(s_2))$$

The final similarity score is then computed as:

$$\text{sim}_{\text{final}} = \text{sim}_{\text{HS}} + \alpha \cdot \text{sim}_{\text{RW}},$$

where $\alpha$ is used as a hyperparameter to control the contribution of RW. To maximize the complementary strengths of HS and RW, we optimize $\alpha$ adaptively at test time. Specifically, $\alpha$ is tuned using a gradient-based approach to maximize the Spearman rank correlation between $\text{sim}_{\text{final}}$ and its components ($\text{sim}_{\text{HS}}$ and $\text{sim}_{\text{RW}}$) over samples for the given task. This process does not require ground truth labels, focusing instead on enhancing complementarity between HS and RW. Once optimized, $\alpha$ remains consistent for the given task and is applied uniformly during testing.

Finally, we compute the rank correlation (e.g., Spearman's rank correlation) between the predicted similarity scores $\text{sim}_{\text{final}}$ and the ground truth similarity. This framework can be applied consistently across other tasks, adapting the weighted sum to task-specific needs.

## 4 EXPERIMENTS

### 4.1 EVALUATION SETUP

We evaluate MoEE on 6 task categories from the MTEB, including Classification, Clustering, Pair Classification, Re-ranking, Semantic Textual Similarity (STS), and Summarization. We focus on

Sentence-to-Sentence (S2S) tasks as they provide a direct and widely-used benchmark for embedding quality. Multilingual datasets are excluded since most MoE LLMs are trained primarily on English data. To control computational costs, we include all STS and Summarization tasks and select tasks with manageable sample sizes from other categories: top-3 for Re-ranking and Classification, top-2 for Clustering, and top-2 for Pair Classification, based on task statistics in Muennighoff et al. (2022). For consistent and fair comparisons, we adopt the MTEB evaluation framework and use task-specific metrics: Accuracy (Classification), V-Measure (Clustering), Average Precision (Pair Classification), Mean Average Precision (Re-ranking), nDCG (Retrieval), and Spearman's correlation (STS and Summarization).

Our experiments use three MoE models:

- DeepSeekMoE-16B (Dai et al., 2024): 28 layers, with 64 experts per layer.
- Qwen1.5-MoE-A2.7B (Team, 2024): 24 layers, each containing 60 experts.
- OLMoE-1B-7B (Muennighoff et al., 2024a): 16 layers, with 64 experts per layer.

All models use per-token routing, but MOEE uses the last token's routing weights, which consistently outperform averaging across all tokens. For the hidden state (HS) embeddings, we use the last-layer hidden state of the last token. Ablation studies supporting this choice are provided in Section 4.3.

**Baselines** Our goal is to extract advanced embedding from MoE LLMs by combining hidden state (HS) and routing weights (RW) *without further training*. To demonstrate the effectiveness of MOEE, we compare it against both RW and HS individually, as well as to several self-supervised and supervised methods that require training. We also assess performance across different prompt strategies, specifically comparing methods without prompts and with PromptEOL (Jiang et al., 2023). For an in-depth analysis of the impact of sequence length on PromptEOL embedding quality, please refer to Appendix B.

Table 3: Performance across MTEB Tasks *without prompts*, including Classification (CLF), Clustering (Clust.), Pair Classification (Pair CLF), Re-ranking (Rerank), STS, and Summarization (Summ.).

| MTEB Tasks | CLF | Clust. | PairCLF | Rerank | STS | Summ. | Avg. |
|---|---|---|---|---|---|---|---|
| DeepSeekMoE-16B | | | | | | | |
| Hidden State (HS) | 44.79 | 25.87 | 44.34 | 38.13 | 34.54 | 24.51 | 35.36 |
| Routing Weight (RW) | 44.06 | 17.53 | 50.59 | 35.94 | 41.11 | 26.22 | 35.91 |
| MOEE (concat) | 44.93 | 24.15 | 51.88 | 41.20 | 46.82 | **31.17** | 40.03 |
| MOEE (sum) | **48.74** | **32.83** | **52.12** | **47.88** | **48.34** | 29.89 | **43.30** |
| Qwen1.5-MoE-A2.7B | | | | | | | |
| Hidden State (HS) | 46.41 | 24.31 | 44.43 | 44.91 | 28.36 | 22.65 | 35.18 |
| Routing Weight (RW) | 38.99 | 10.55 | 42.26 | 33.53 | 23.97 | 27.44 | 29.46 |
| MOEE (concat) | 44.81 | 26.75 | 49.79 | 49.23 | 37.93 | **27.61** | 39.35 |
| MOEE (sum) | **50.70** | **31.35** | **51.87** | **49.82** | **45.75** | 24.00 | **42.25** |
| OLMoE-1B-7B | | | | | | | |
| Hidden State (HS) | 44.23 | 23.79 | 47.56 | 45.60 | 35.44 | 20.94 | 36.26 |
| Routing Weight (RW) | 43.54 | 17.66 | **53.12** | 40.91 | 44.68 | 28.68 | 38.10 |
| MOEE (concat) | 44.62 | 22.83 | 51.64 | 46.58 | 48.84 | **31.67** | 41.03 |
| MOEE (sum) | **48.54** | **30.67** | 50.93 | **47.77** | **49.45** | 28.77 | **42.69** |

## 4.2 MAIN RESULTS

Our method demonstrates consistent performance improvements across a variety of MTEB tasks, as shown in Tables 3 and 4. Results for datasets under each task type are detailed in Appendix A. MOEE that combines routing weights with hidden state consistently outperforms both standalone methods (RW and HS) in most cases, highlighting the complementary nature of these two components.

For tasks evaluated without prompts, the results show that MOEE (sum) achieves the highest average performance across models, with notable improvements in tasks such as Classification, Re-ranking, and STS. Specifically, DeepSeekMoE shows a substantial boost from 35.36 (HS) to 43.30 (MOEE (sum)), a 22.45% improvement. This pattern holds across Qwen1.5-MoE and OLMoE, where MOEE (sum) achieves consistent gains over both individual methods. When PromptEOL is introduced (Table 4), we observe even greater performance gains, with 25.96% improvement for DeepSeekMoE.

Table 4: Performance across MTEB Tasks when *PromptEOL* (Jiang et al., 2023) is applied to MoE. Baselines marked with ⋆ are sourced from the MTEB leaderboard (Muennighoff et al., 2022) and require training.

| MTEB Tasks | CLF | Clust. | PairCLF | Rerank | STS | Summ. | Avg. |
|---|---|---|---|---|---|---|---|
| *Self-Supervised Methods* | | | | | | | |
| Glove⋆ (Pennington et al., 2014) | 51.04 | 23.11 | 62.90 | 48.72 | 60.52 | 28.87 | 45.86 |
| Komninos⋆ (Komninos & Manandhar, 2016) | 50.21 | **24.96** | 66.63 | 50.03 | 61.73 | 30.49 | 47.34 |
| BERT⋆ (Devlin, 2018) | 52.36 | 23.48 | 66.10 | 48.47 | 52.89 | 29.82 | 45.52 |
| SimCSE-BERT-unsup⋆ (Gao et al., 2021) | **54.80** | 22.59 | **70.79** | **52.42** | **75.00** | **31.15** | **51.13** |
| *Supervised Methods* | | | | | | | |
| SimCSE-BERT-sup⋆ | **58.98** | 29.49 | **75.82** | 53.61 | **79.97** | 23.31 | 53.53 |
| coCondenser-msmarco⋆ (Gao & Callan, 2021) | 53.89 | **32.85** | 74.56 | **60.08** | 76.41 | 29.50 | **54.55** |
| SPECTER⋆ (Cohan et al., 2020) | 42.59 | 27.94 | 56.24 | 55.87 | 60.68 | 27.66 | 45.16 |
| LaBSE (Feng et al., 2020) | 54.31 | 24.05 | 73.68 | 54.63 | 70.95 | **31.05** | 51.45 |
| LASER2 | 42.54 | 14.01 | 70.52 | 46.99 | 64.52 | 26.80 | 44.23 |
| SGPT-125M-nli (Muennighoff, 2022) | 53.28 | 26.59 | 68.80 | 53.65 | 75.01 | 30.26 | 51.27 |
| *DeepSeekMoE-16B* | | | | | | | |
| Hidden State (HS) | 58.24 | 24.64 | 48.76 | 38.13 | 59.66 | 24.38 | 42.30 |
| Routing Weight (RW) | 49.52 | 19.97 | 68.30 | 37.48 | 59.52 | **29.26** | 44.01 |
| MoEE (concat) | 54.21 | 26.10 | **72.44** | 53.31 | 67.59 | 28.89 | 50.42 |
| MoEE (sum) | **58.31** | **34.52** | 70.95 | **55.99** | **70.66** | 29.22 | **53.28** |
| *Qwen1.5-MoE-A2.7B* | | | | | | | |
| Hidden State (HS) | 59.34 | 29.50 | **74.29** | 56.51 | 67.39 | 23.01 | 51.67 |
| Routing Weight (RW) | 47.84 | 16.74 | 64.85 | 43.55 | 51.71 | 27.74 | 42.07 |
| MoEE (concat) | 54.23 | 27.18 | 73.93 | 56.12 | 68.52 | 28.57 | 51.43 |
| MoEE (sum) | **59.57** | **38.33** | 72.21 | **56.25** | **72.78** | **31.09** | **55.04** |
| *OLMoE-1B-7B* | | | | | | | |
| Hidden State (HS) | **58.18** | 32.83 | **72.10** | 58.31 | 72.91 | 27.96 | 53.72 |
| Routing Weight (RW) | 45.02 | 19.93 | 61.58 | 43.91 | 54.33 | 29.49 | 42.38 |
| MoEE (concat) | 52.59 | 33.92 | 71.85 | 56.69 | 71.13 | 30.21 | 52.73 |
| MoEE (sum) | 57.46 | **36.46** | 71.26 | **60.43** | **74.63** | 30.71 | **55.16** |

Across all models, MoEE (sum) again leads to the best results, with OLMoE achieving the highest overall average of 55.16 and Qwen1.5-MoE following closely at 55.04. While MoEE shows marginal gains over HS in the Classification task, this is expected, as the final layer HS is more aligned with output-specific features, which benefits classification.

Although MoEE initially trails behind self-supervised and supervised methods without prompts, the introduction of PromptEOL leads to a significant shift. As shown in Table 4, MoEE surpasses supervised approaches like SimCSE and coCondenser, achieving superior performance without requiring additional training. This underscores both its effectiveness and efficiency.

## 4.3 Ablation Study

Table 5: Ablation study on different ways of using routing weights (RW) and hidden state (HS).

| STS Datasets | STS12 | STS13 | STS14 | STS15 | STS16 | Avg. |
|---|---|---|---|---|---|---|
| *DeepSeekMoE-16B* | | | | | | |
| HS - last token, last layer | 51.99 | **69.56** | **54.68** | **58.04** | **68.47** | **60.40** |
| HS - last token, all layers | 59.82 | 60.59 | 45.20 | 51.08 | 58.88 | 55.03 |
| HS - all tokens, last layer | 30.95 | 34.42 | 26.77 | 34.90 | 37.11 | 32.78 |
| HS - all tokens, all layers | **60.81** | 62.46 | 46.90 | 52.38 | 59.99 | 56.34 |
| RW - last token | **61.97** | **65.86** | **51.38** | **65.86** | **62.49** | **61.18** |
| RW - all tokens | 50.76 | 46.42 | 41.47 | 43.68 | 48.37 | 46.03 |
| MoEE (best) | 67.39 | 81.43 | 68.98 | 67.76 | 74.26 | 71.75 |

This ablation study investigates how different methods of extracting routing weights (RW) and hidden state (HS) affect embedding quality across the STS12-16 datasets, with results presented in Table 5.

As detailed in Section 3.1, RW integrates routing decisions across all layers, capturing information at multiple depths. In contrast, HS from only the last layer may miss important intermediate details. Therefore, we evaluate the use of hidden states from all layers (*HS - last token, all layers*) to see if it can match RW, which naturally leverages multi-layered information.

We also assess the impact of using only the last token versus averaging across all tokens. While the last token often condenses crucial sequence information, mean pooling across all tokens may offer a broader view by incorporating contributions from every token. Thus, we compare *HS - last token* with *HS - all tokens*, and *RW - last token* with *RW - all tokens*. For multi-layer or multi-token cases, mean pooling is applied.

Our results show that focusing on the last token, whether from HS or RW, consistently delivers the best performance. This indicates that the last token captures the most critical semantic information, while pooling across tokens or layers introduces noise. Notably, RW outperforms HS, underscoring its superior ability to capture nuanced, dynamic information that HS alone cannot replicate.

### 4.4 A Stability Comparison of RW and HS using different Prompts

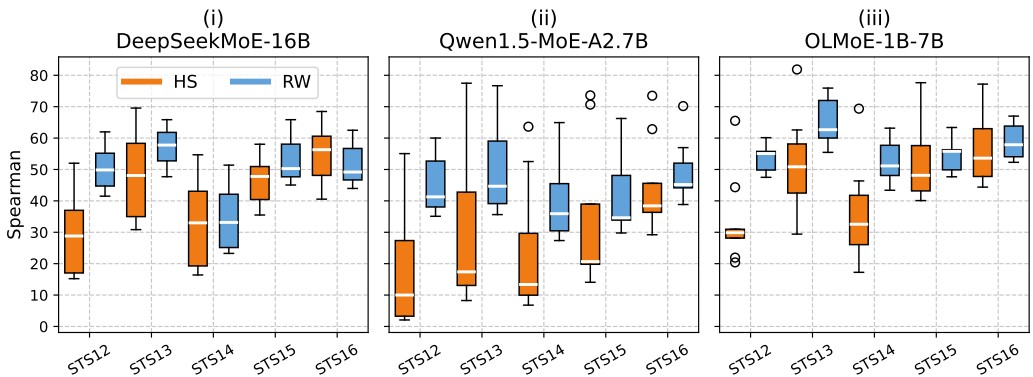

Figure 5: Box plots of the performance of the two embedding methods (RW or HS) using nine different prompts (listed in Table 2) on five STS datasets, evaluated on three MoE models: (i) DeepSeekMoE, (ii) Qwen1.5-MoE, and (iii) OLMoE. The higher variance and wider spread of HS in the box plots indicate its sensitivity to the prompt choice, while RW is more robust (lower variance) with better mean performance.

Prompts are commonly used to boost the performance of embedding models across diverse downstream tasks (Lei et al., 2024), as shown by the improved results of PromptEOL (Table 3) compared to no prompts (Table 4). However, the effectiveness of these prompts can vary, and a method's robustness depends on its ability to handle these variations. To assess the prompt sensitivity of RW and HS, we measure their Spearman correlation scores across STS12-16 datasets using 9 different prompts listed in Table 2. The analysis is performed on three MoE models: DeepSeekMoE, Qwen1.5-MoE, and OLMoE, which differ in model size and architecture, allowing us to investigate the generalizability of our findings. For each model, we compute the mean and variance of these scores for each dataset, capturing how performance fluctuates under different prompt conditions and whether the methods remain stable when exposed to prompt variations.

Figure 5 highlights the performance variance for both methods. **HS exhibits significantly higher variance**, indicating that its performance is highly dependent on the specific prompt used. This suggests that **HS is more sensitive to prompt formulation**, leading to inconsistent results that could hinder its reliability in broader applications. Figure 4 (see Section 3.2) further supports this from another perspective[3], showing a smaller mean correlation of 0.52 for HS using different prompts, reflecting a higher variance than RW.

In contrast, **RW demonstrates greater stability**, with consistently lower variance and narrower box plots across all datasets, indicating its **robustness to prompt choice**. In Figure 4, RW also achieves

---

[3]The Spearman correlation in Figure 5, as a performance metric, is between HS/RW and the ground truth, while the Spearman correlation in Figure 4 is to compare different embedding.

a higher mean correlation of 0.63 between different prompts, underscoring its ability to maintain stable performance across different prompts. This makes MOEE a more reliable option for tasks where prompt variability is expected.

Notably, Qwen1.5-MoE and OLMoE exhibit greater sensitivity to prompt variations compared to DeepSeekMoE. Despite this, HS embeddings consistently demonstrate significantly higher variance than RW embeddings across all tasks and models. This pattern of RW robustness holds consistently across the three MoE models, reinforcing its stability in diverse settings.

## 4.5 CASE STUDY: WHEN HS OUTPERFORMS RW & VICE VERSA

Table 6: Semantically similar sentence pairs correctly predicted by HS embedding but not by RW embedding. Differences between the sentences are highlighted to show subtle variations that influence prediction outcomes.

| | Sentence 1 | Sentence 2 |
|---|---|---|
| 1 | the vote will take place today at 5.30 p.m | the vote will take place at **17h30** |
| 2 | the standards are scarcely comparable, let alone transferable | the **norms** are **hardly** comparable and **still less** transferable |
| 3 | that provision could open the door wide to arbitrariness | this **point of procedure** opens the door to the **arbitrary** |
| 4 | A woman puts flour on a piece of meat | A woman **is putting** flour **onto some** meat. |
| 5 | the fishermen are inactive, tired and disappointed | fishermen are inactive, tired and **disappointment** |

Table 7: Semantically similar sentence pairs correctly predicted by RW embedding but not by HS embedding.

| | Sentence 1 | Sentence 2 |
|---|---|---|
| 1 | He did, but the initiative did not get very far. | What happened is that the initiative does not go very far. |
| 2 | then perhaps we could have avoided a catastrophe | we might have been able to prevent a disaster |
| 3 | it increases the power of the big countries at the expense of the small countries | it has the effect of augmenting the potency of the big countries to the detriment of babies |
| 4 | festive social event, celebration | an occasion on which people can assemble for social interaction and entertainment. |
| 5 | group of people defined by a specific profession | organization of performers and associated personnel (especially theatrical). |

In this section, we analyze instances where HS embedding performs better than RW embedding (Table 6), as well as cases where RW outperforms HS (Table 7). This helps identify the strengths and weaknesses of each method and offers insights into when one may be preferred over the other.

From the results, **HS embeddings excel in capturing formal linguistic consistency**, particularly when sentence structure **undergoes only superficial changes**. They effectively represent the overall structure and meaning of sentences, making them useful in cases with minimal semantic variation. In contrast **RW embedding performs better when handling paraphrasing, synonym use, and nuanced stylistic shifts**. The RW mechanism's sensitivity to input variations allows it to capture deeper contextual changes, even when the overall meaning of the sentence is preserved.

## 5 CONCLUSION

In this paper, we explore the untapped potential of MoE as effective embedding generators without extra training. Our analysis reveals that RW derived from MoE models complements the widely-used HS embedding, offering a deeper understanding of input semantics. By leveraging both RW and HS, we propose MOEE, which significantly improves embedding performance across diverse tasks in the MTEB benchmark. Our results demonstrate that combining RW and HS boosts generalization, making MoE models versatile tools for embedding tasks. Future work would further explore how to leverage MOEE adaptively for task-specific scenarios.

## ACKNOWLEDGMENTS

We would like to thank Rocco Zhang for his preliminary attempts and experiments in the earlier-stage exploration of this project's initial idea. We appreciate the reviewers and area chairs for their insightful comments and suggestions.

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

# A   MTEB RESULTS

We present detailed evaluation results of task types, including STS (Table 8), classification (Table 9), pair classification (Table 10), clustering (Table 11), and re-ranking (Table 12) tasks. We show the performance of our method across different models and prompts, and compares it to baseline methods like Hidden State (HS) and Routing Weight (RW).

Table 8: Detailed Results of STS Tasks. The DeepSeekMoE, Qwen1.5-MoE, and OLMoE models are evaluated on tasks from STS12 to STSBenchmark. The MoEE method (without and with PromptEOL) significantly improves performance across most benchmarks.

| | Prompt | STS12 | STS13 | STS14 | STS15 | STS16 | BIOSSES | SICK-R | STSBenchmark |
|---|---|---|---|---|---|---|---|---|---|
| **DeepSeekMoE-16b** | | | | | | | | | |
| Hidden State (HS) | none | 20.90 | 43.39 | 24.02 | 37.75 | 47.15 | 29.87 | 42.66 | 30.61 |
| Routing Weight (RW) | none | 45.22 | 41.38 | 28.75 | 38.63 | 50.36 | 34.14 | 51.98 | 38.44 |
| MoEE (concat) | none | 46.26 | 55.88 | 37.90 | 42.37 | 54.19 | 41.20 | 53.66 | 43.06 |
| MoEE (sum) | none | **46.41** | **60.58** | **41.50** | **42.85** | **54.98** | **42.33** | **53.70** | **44.36** |
| Hidden State (HS) | PromptEOL | 51.99 | 69.56 | 54.68 | 58.04 | 68.47 | 45.29 | 63.78 | 65.48 |
| Routing Weight (RW) | PromptEOL | 61.97 | 65.86 | 51.38 | 65.86 | 62.49 | 53.97 | 57.93 | 56.68 |
| MoEE (concat) | PromptEOL | 66.79 | 77.60 | 63.56 | 64.60 | 71.22 | 61.96 | 66.29 | 68.72 |
| MoEE (sum) | PromptEOL | **67.39** | **81.43** | **68.98** | **67.76** | **74.26** | **62.09** | **69.98** | **73.41** |
| **Qwen1.5-MoE-A2.7B** | | | | | | | | | |
| Hidden State (HS) | none | 8.39 | 25.23 | 15.76 | 22.08 | 38.11 | 28.69 | 51.73 | 36.88 |
| Routing Weight (RW) | none | 27.96 | 18.89 | 13.88 | 17.11 | 36.29 | 25.40 | 29.42 | 22.80 |
| MoEE (concat) | none | 33.36 | 36.30 | 24.68 | 25.86 | 47.16 | 39.06 | 53.92 | 43.09 |
| MoEE (sum) | none | **35.72** | **47.29** | **31.51** | **31.00** | **50.61** | **53.40** | **62.35** | **54.11** |
| Hidden State (HS) | PromptEOL | 55.05 | 77.48 | 63.63 | **73.60** | 73.49 | 61.42 | 67.01 | 67.42 |
| Routing Weight (RW) | PromptEOL | 54.39 | 59.05 | 45.49 | 48.11 | 56.96 | 43.65 | 55.46 | 50.54 |
| MoEE (concat) | PromptEOL | 64.44 | 77.38 | 64.05 | 67.18 | 71.48 | 64.87 | 69.01 | 69.71 |
| MoEE (sum) | PromptEOL | **65.54** | **82.44** | **71.39** | 72.88 | **75.43** | **67.84** | **71.15** | **75.57** |
| **OLMoE-1B-7B** | | | | | | | | | |
| Hidden State (HS) | none | 21.53 | 41.47 | 22.71 | 39.88 | 51.49 | 44.11 | 39.98 | 22.36 |
| Routing Weight (RW) | none | 47.16 | 43.92 | 32.62 | 43.87 | 51.91 | 44.30 | 52.89 | 40.77 |
| MoEE (concat) | none | 48.82 | 52.69 | 37.48 | 46.80 | 56.06 | **54.58** | 52.24 | 42.02 |
| MoEE (sum) | none | **49.59** | **54.19** | **38.87** | **47.27** | **56.11** | **54.58** | **52.82** | **42.16** |
| Hidden State (HS) | PromptEOL | 65.51 | 81.86 | 69.37 | **77.64** | 77.19 | 73.54 | 66.62 | 71.51 |
| Routing Weight (RW) | PromptEOL | 55.76 | 60.01 | 48.08 | 49.88 | 57.88 | 56.28 | 56.02 | 50.72 |
| MoEE (concat) | PromptEOL | 67.35 | 80.13 | 68.42 | 68.76 | 73.35 | 73.02 | 67.51 | 70.47 |
| MoEE (sum) | PromptEOL | **68.84** | **84.34** | **74.02** | 73.81 | 76.88 | 73.02 | **70.56** | **75.58** |

Table 9: Detailed Results of Classification Tasks, including sentiment extraction, emotion classification, and toxic conversations classification. The performance of different methods (Hidden State, Routing Weight, and MOEE) with and without PromptEOL is shown.

| | Prompt | TweetSentiment-Extraction-Classification | Emotion-Classification | Toxic-Conversations-Classification |
|---|---|---|---|---|
| **DeepSeekMoE-16b** | | | | |
| Hidden State (HS) | none | 49.14 | 27.55 | 57.69 |
| Routing Weight (RW) | none | 52.37 | 26.49 | 53.32 |
| MOEE (concat) | none | **52.64** | **28.02** | 54.12 |
| MOEE (sum) | none | 50.32 | 27.52 | **68.39** |
| Hidden State (HS) | PromptEOL | 60.13 | **49.11** | 65.47 |
| Routing Weight (RW) | PromptEOL | 57.68 | 35.57 | 55.32 |
| MOEE (concat) | PromptEOL | **61.12** | 45.59 | 55.93 |
| MOEE (sum) | PromptEOL | 59.32 | 46.86 | **68.76** |
| **Qwen1.5-MoE-A2.7B** | | | | |
| Hidden State (HS) | none | 48.83 | 31.02 | 59.38 |
| Routing Weight (RW) | none | 42.80 | 20.63 | 53.53 |
| MOEE (concat) | none | **49.60** | 30.93 | 53.90 |
| MOEE (sum) | none | 48.84 | **32.76** | **70.50** |
| Hidden State (HS) | PromptEOL | **61.14** | **48.09** | 68.80 |
| Routing Weight (RW) | PromptEOL | 55.33 | 33.82 | 54.37 |
| MOEE (concat) | PromptEOL | 60.78 | 46.10 | 55.82 |
| MOEE (sum) | PromptEOL | 60.72 | 47.97 | **70.03** |
| **OLMoE-1B-7B** | | | | |
| Hidden State (HS) | none | 50.29 | **30.29** | 52.10 |
| Routing Weight (RW) | none | 50.15 | 25.53 | 54.93 |
| MOEE (concat) | none | **51.59** | 28.76 | 53.51 |
| MOEE (sum) | none | 51.00 | 29.75 | 64.86 |
| Hidden State (HS) | PromptEOL | 59.58 | **47.50** | **67.46** |
| Routing Weight (RW) | PromptEOL | 52.79 | 28.51 | 53.75 |
| MOEE (concat) | PromptEOL | 59.72 | 42.78 | 55.27 |
| MOEE (sum) | PromptEOL | **59.92** | 45.63 | 66.84 |

Table 10: Pair classification task results on TwitterURLCorpus and TwitterSemEval2015.

| | Prompt | TwitterURLCorpus | TwitterSemEval2015 |
|---|---|---|---|
| **DeepSeekMoE-16b** | | | |
| Hidden State (HS) | none | 49.04 | 39.63 |
| Routing Weight (RW) | none | 53.39 | **47.79** |
| MOEE (concat) | none | 57.27 | 46.48 |
| MOEE (sum) | none | **58.99** | 45.25 |
| Hidden State (HS) | PromptEOL | 36.72 | 60.79 |
| Routing Weight (RW) | PromptEOL | 76.58 | 60.01 |
| MOEE (concat) | PromptEOL | **80.08** | **64.79** |
| MOEE (sum) | PromptEOL | 79.20 | 62.70 |
| **Qwen1.5-MoE-A2.7B** | | | |
| Hidden State (HS) | none | 45.71 | 43.14 |
| Routing Weight (RW) | none | 48.78 | 35.74 |
| MOEE (concat) | none | 53.74 | 45.83 |
| MOEE (sum) | none | **57.78** | **45.95** |
| Hidden State (HS) | PromptEOL | **82.50** | **66.07** |
| Routing Weight (RW) | PromptEOL | 73.72 | 55.98 |
| MOEE (concat) | PromptEOL | 82.34 | 65.51 |
| MOEE (sum) | PromptEOL | 80.21 | 64.20 |
| **OLMoE-1B-7B** | | | |
| Hidden State (HS) | none | 55.07 | 40.04 |
| Routing Weight (RW) | none | 54.25 | **51.99** |
| MOEE (concat) | none | 56.97 | 46.31 |
| MOEE (sum) | none | **57.03** | 44.82 |
| Hidden State (HS) | PromptEOL | **82.32** | **61.87** |
| Routing Weight (RW) | PromptEOL | 70.37 | 52.79 |
| MOEE (concat) | PromptEOL | **82.32** | 61.38 |
| MOEE (sum) | PromptEOL | 80.98 | 61.53 |

Table 11: Clustering task results, showing performance on TwentyNewsgroupsClustering and Medrx-ivClusteringS2S. MOEE (sum) consistently performs best without a prompt, while the MOEE method with PromptEOL delivers substantial gains.

| | Prompt | TwentyNewsgroupsClustering | MedrxivClusteringS2S |
|---|---|---|---|
| **DeepSeekMoE-16b** | | | |
| Hidden State (HS) | none | 25.62 | 26.11 |
| Routing Weight (RW) | none | 15.33 | 19.72 |
| MOEE (concat) | none | 22.94 | 25.35 |
| MOEE (sum) | none | **31.44** | **34.22** |
| Hidden State (HS) | PromptEOL | 27.02 | 22.26 |
| Routing Weight (RW) | PromptEOL | 21.89 | 18.04 |
| MOEE (concat) | PromptEOL | 29.13 | 23.06 |
| MOEE (sum) | PromptEOL | **35.77** | **33.27** |
| **Qwen1.5-MoE-A2.7B** | | | |
| Hidden State (HS) | none | 26.14 | 22.48 |
| Routing Weight (RW) | none | 9.71 | 11.38 |
| MOEE (concat) | none | 28.99 | 24.51 |
| MOEE (sum) | none | **32.07** | **30.62** |
| Hidden State (HS) | PromptEOL | 34.04 | 24.95 |
| Routing Weight (RW) | PromptEOL | 16.94 | 16.54 |
| MOEE (concat) | PromptEOL | 30.45 | 23.91 |
| MOEE (sum) | PromptEOL | **42.05** | **34.60** |
| **OLMoE-1B-7B** | | | |
| Hidden State (HS) | none | 21.05 | 26.52 |
| Routing Weight (RW) | none | 17.14 | 18.17 |
| MOEE (concat) | none | 20.72 | 24.94 |
| MOEE (sum) | none | **27.58** | **33.75** |
| Hidden State (HS) | PromptEOL | 38.96 | 26.69 |
| Routing Weight (RW) | PromptEOL | 22.13 | 17.72 |
| MOEE (concat) | PromptEOL | **41.23** | 26.60 |
| MOEE (sum) | PromptEOL | 38.58 | **34.33** |

Table 12: Re-ranking task results, showing performance on AskUbuntu, SciDocsRR, and StackOver-flow duplicate questions re-ranking tasks.

| | Prompt | AskUbuntuDupQuestions | SciDocsRR | StackOverflowDupQuestions |
|---|---|---|---|---|
| **DeepSeekMoE-16b** | | | | |
| Hidden State (HS) | none | 43.75 | 45.23 | 25.79 |
| Routing Weight (RW) | none | 41.97 | 42.65 | 23.21 |
| MOEE (concat) | none | 44.10 | 53.43 | 26.06 |
| MOEE (sum) | none | **45.26** | **70.79** | **27.58** |
| Hidden State (HS) | PromptEOL | 43.75 | 45.23 | 25.41 |
| Routing Weight (RW) | PromptEOL | 46.57 | 42.65 | 23.21 |
| MOEE (concat) | PromptEOL | 50.66 | 72.63 | 36.65 |
| MOEE (sum) | PromptEOL | **52.93** | **76.17** | **38.88** |
| **Qwen1.5-MoE-A2.7B** | | | | |
| Hidden State (HS) | none | 43.71 | 60.91 | 30.12 |
| Routing Weight (RW) | none | 41.00 | 36.85 | 22.75 |
| MOEE (concat) | none | **44.95** | 68.42 | **34.31** |
| MOEE (sum) | none | 44.30 | **70.85** | **34.31** |
| Hidden State (HS) | PromptEOL | **54.69** | 75.06 | 39.79 |
| Routing Weight (RW) | PromptEOL | 44.65 | 55.03 | 30.96 |
| MOEE (concat) | PromptEOL | 52.15 | **75.69** | 40.51 |
| MOEE (sum) | PromptEOL | 51.30 | 74.53 | **42.91** |
| **OLMoE-1B-7B** | | | | |
| Hidden State (HS) | none | 43.67 | 69.08 | 24.05 |
| Routing Weight (RW) | none | 42.83 | 54.17 | 25.72 |
| MOEE (concat) | none | 43.91 | 70.33 | 25.49 |
| MOEE (sum) | none | **44.57** | **72.54** | **26.20** |
| Hidden State (HS) | PromptEOL | 55.32 | 78.24 | 41.36 |
| Routing Weight (RW) | PromptEOL | 45.11 | 55.43 | 31.20 |
| MOEE (concat) | PromptEOL | 52.81 | 77.14 | 40.13 |
| MOEE (sum) | PromptEOL | **56.68** | **81.19** | **43.41** |

# B ANALYSIS: IMPACT OF TEXT LENGTH ON PROMPTEOL EMBEDDING QUALITY

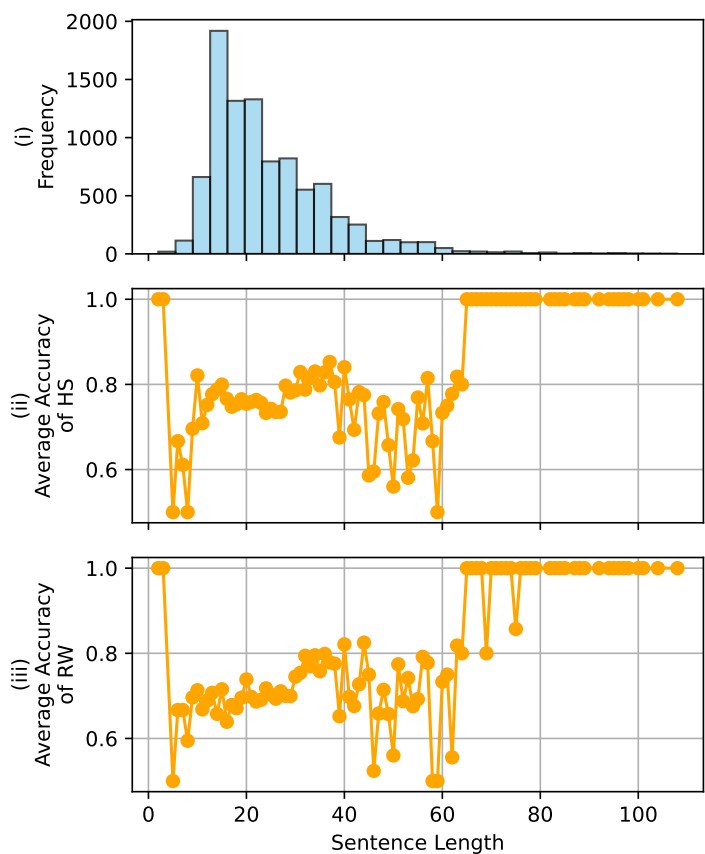

Figure 6: (i) Distribution of sentence lengths in the STS12-16 dataset, concentrated on short to medium-length sequences. (ii) and (iii) show that sentence length has no significant negative impact on the average accuracy of Hidden State (HS) and Routing Weights (RW) embeddings, respectively.

PromptEOL is designed to condense the meaning of a sentence into a single word, making it well-suited for shorter sequences. However, its robustness for longer or more complex inputs raises important questions. Specifically, does sequence length affect the quality of the embeddings? Addressing this is crucial for evaluating PromptEOL's effectiveness and identifying areas for improvement.

This analysis leverages the STS12-16 datasets, which provide diverse sentence pairs commonly used in Sentence-to-Sentence (S2S) tasks. These datasets offer a representative sample of real-world sentence lengths, making them ideal for studying the relationship between length and embedding quality. We first examine the length distribution to understand the typical input range and then analyze how sequence length correlates with embedding quality, measured by accuracy. Accuracy is determined by how closely the embedding-based ranking aligns with the ground truth, with deviations beyond a threshold considered correct.

Figure 6 (i) illustrates the length distribution in the STS12-16 datasets, with a concentration of short to medium-length sequences and a median of approximately 25 tokens. Figure 6 (ii) and (iii) shows the relationship between sentence length and average accuracy for HS and RW embeddings generated using PromptEOL, respectively. The results indicate no significant negative correlation, confirming that sentence length does not adversely affect embedding quality within the S2S context for both Hidden State (HS) and Routing Weights (RW). These findings demonstrate that PromptEOL effectively captures the semantic meaning of shorter sequences without being sensitive to variations in length.

However, for tasks involving longer sequences, such as Paragraph-to-Paragraph (P2P) embeddings, challenges may emerge. Compressing extensive information into a single word may limit embedding quality. Addressing this limitation—for instance, by segmenting longer texts or introducing multi-token sinks—remains an important direction for future work.

