# OpenReview forum: "Your Mixture-of-Experts LLM Is Secretly an Embedding Model for Free"
_ICLR.cc/2025/Conference — ICLR 2025 Oral_

### Official Review · Reviewer_rfVb · 2024-10-31

**Soundness:** 3
**Presentation:** 3
**Contribution:** 3
**Rating:** 8
**Confidence:** 4

**Summary:**

The paper explores the use of the routing weights of a mixture-of-expert LLM for embedding tasks. Concatenating those routing weights over layers gives them surprisingly good scores. The paper claims that the routing weights are more robust to varying prompts than the contextualized representations and that the routing weights complement the standard contextualized representations well for embedding-centric downstream tasks. The experimental part covers tasks from the MTEB benchmark, e.g., on textual similarity, classification, and clustering. The results shows that the combination of routing weights with hidden states are preferable over the current best-practice of just using the hidden states of language models for embedding-centric tasks.

**Strengths:**

The idea is original, and the experimental results confirm its effectiveness. The MoEE technique (essentially combining hidden states with routing weights) scores higher in 5 out of 6 tasks and gives equal performance on the 6th task. This hints at the approach of making use of MoE routing weights could be generally considered for embedding-centric tasks. The paper thoroughly tests the contribution of hidden states and routing weights to the overall performance (Sec 3.2). The paper further explores different variants of dealing with multiple sequences, i.e. concatenation or calculating and interpolating cosine similarity separately for hidden states and routing weights. Here the results show that the sum variant is in most cases preferable. Generally, a strength of the approach is that no further training is required, but that those embeddings just need to be extracted from a language model. The paper further presents an ablation study comparing different token/layer selection strategies for the hidden states and routing weights -- interestingly here, routing weights already perform better than the hidden states.

**Weaknesses:**

To some extent the paper relies PromptEOL, i.e. prompting the model to distill the meaning of a sentence into one word. While this is clearly stated in the paper already, it could potentially hint at further limitations; for example, when a large context would need to be compressed into a single word.

Although the baseline selection is solid, a few more commonly used models for embedding-centric tasks could be employed. SentenceBERT, ColBERT, & BertScore come to my mind, as those are commonly employed for text similarity and such tasks.

**Questions:**

Do you have any insights how the length of the sequence has an influence on embedding quality?

Given last-layer last-token HS performs best in the ablations, I assume that is what you use for the main table. Is this correct or did I miss anything?

---

> ### Author Response · Authors · 2024-11-21
> **Response to Reviewer rfVb**
>
> Dear Reviewer rfVb,
>
> Thank you for your insightful review and for highlighting the strengths of our work. We are pleased to provide responses and clarifications to address the points you raised.
>
> ---
>
> > Q1. **"To some extent the paper relies PromptEOL, i.e. prompting the model to distill the meaning of a sentence into one word. While this is clearly stated in the paper already, it could potentially hint at further limitations. Do you have any insights how the length of the sequence has an influence on embedding quality?"**
>
> Thank you for raising this insightful point. Our primary focus is on sentence-to-sentence (S2S) tasks, where sequence lengths are generally short. As detailed in Appendix B, an analysis of STS12-16 samples shows no evidence of a negative correlation between sentence length and performance for both Hidden State (HS) and Routing Weights (RW) embeddings. This demonstrates that PromptEOL effectively captures the essence of relatively short sequences without being impacted by length variations.
>
> For longer contexts, such as paragraph-to-paragraph (P2P) tasks, we acknowledge that compressing extensive information into a single word may be too aggressive. To address this, we can split longer paragraphs into smaller sentences, generating a "one word" embedding for each, and combining them hierarchically. Another promising approach is sampling multiple sink tokens within a single context to capture diverse aspects of the input. These strategies would enhance PromptEOL's ability to handle longer sequences effectively.
>
>
> > Q2. **"A few more commonly used models for embedding-centric tasks could be employed."**
>
>
> Thank you for the suggestion. To strengthen our analysis, we have added more baselines compatible with the MTEB evaluation framework, including the Language-agnostic BERT Sentence Encoder (LaBSE) [1], in addition to BERTScore, which was already included.
>
> It is worth noting that our aim is not to directly compare against state-of-the-art embedding models but rather to investigate the potential of autoregression trained decoder-only LLMs as an encoder. We demonstrate the potential of **Mixture-of-Experts (MoE) LLMs** to generate high-quality embeddings **for free**—sometimes even surpassing models specifically trained for embedding tasks, such as SimCSE and coCondenser. As more MoE LLMs become available excelling on generation tasks, extracting training-free embedding from them becomes increasingly intriguing and promising.
>
>
> > Q3. **"Given last-layer last-token HS performs best in the ablations, I assume that is what you use for the main table. Is this correct or did I miss anything?"**
>
> You are correct; the last-layer, last-token HS configuration, which showed the most robust results in our ablation studies, is used in the main experimental table. We have updated the experimental setup section to clarify this.
>
> ---
>
> Thank you once again for your valuable feedback and thorough assessment. We have updated the manuscript and highlighted the changes in blue for your convenience.
>
> Best regards,
> The Authors
>
>
> [1] Feng, Fangxiaoyu, et al. "Language-agnostic BERT sentence embedding." arXiv preprint arXiv:2007.01852 (2020).

---

> > ### Comment · Reviewer_rfVb · 2024-11-23
> >
> > Thank you for the clarification and the discussion on sequence length. I suggest to link to Appendix B at an appropriate point in the main text.

---

> > > ### Author Response · Authors · 2024-11-23
> > > **Follow-up**
> > >
> > > Thank you for your thoughtful suggestion! We have included a reference to Appendix B in Section 4.1 of the revised version for clarity and completeness.

---

### Official Review · Reviewer_A92b · 2024-11-07

**Soundness:** 3
**Presentation:** 3
**Contribution:** 2
**Rating:** 6
**Confidence:** 3

**Summary:**

This paper first shows that RW and HS of MoE LLM are complementary for embedding-focused tasks. And they proposed two method to combine these two representations, concat and sum respectively. They benchmark the performance of the method on 6 embedding tasks of MTEB and show that it can outperform both RW and HS, as well as other previous baselines. In addition, they demonstrate the robustness of the method to different prompts.

**Strengths:**

1. This paper explores the utility of the RW and HS components of MoE LLMs for embedding-focused tasks, an area that has not yet been explored by others.
2. They have comprehensive experiment on 6 embedding tasks of MTEB with three different MoE LLMs and the method is stable across different prompts.
3. The logic is easy to follow and understand.

**Weaknesses:**

1. Although the authors have compared their method to previous approaches, the baselines used are somewhat outdated. For instance, the latest baseline of this paper is published in 2021. This a old topic or authors just missed some latest baselines ( I think this could limit the contribution of this paper to the field.)?
2. The method seems not sensitive to the scale of MoE LLM, could author provide some explanation?
3. Though authors demonstrate the stability of the method compared to HS, it is not comprehensive without stability of baselines in the main experiment.
4. It seems the performant variant of their method "sum". And "sum" depends on the hyper parameter alpha. Could authors provide more details of how to tune alpha? Is alpha consistent across all experiments?

**Questions:**

1. Could author provide more recent baselines, cause the best baseline in this paper from 2021?
2. Could author explain their method is not sensitive to the scale of models?
3. Could authors also report the stability of baselines from the main experiments?
4. Could authors provide more details about how to tune alpha and is it consistent across experimentation?

---

> ### Author Response · Authors · 2024-11-21
> **Response to Reviewer A92b**
>
> Dear Reviewer A92b,
>
> Thank you for your thoughtful review and for recognizing the strengths of our work. We are pleased to address your questions and suggestions below.
>
> ---
>
> > **Q1. "The baselines used are somewhat outdated. For instance, the latest baseline of this paper is published in 2021. Could author provide more recent baselines?"**
>
> We appreciate this suggestion. Our focus is not on comparing against state-of-the-art, which are models specifically trained for embedding tasks, but rather on investigating the capability of auto-regression pre-trained decoder LLM in producing effective embedding (i.e., using decoder as an encoder). This is critical to unifying the two types of models.
>
> Our empirical results are encouraging: for the first time, we demonstrated that effective **training-free embedding** can be achieved by leveraging **Mixture-of-Experts (MoE) LLMs**, which can even surpass training-based embedding by SimCSE or coCondenser. Our primary baselines are still training-free methods, such as **PromptEOL (2023)** [1], which align better with the goal of developing scalable and resource-efficient embedding approaches. As per your suggestion, we have added new baselines compatible with the MTEB evaluation framework in the revised manuscript.
>
> > **Q2. "The method seems not sensitive to the scale of MoE LLM. Could author provide some explanation?"**
>
> Scaling laws are typically observed more evidently on the same architecture. However, the three MoE models in our experiments vary significantly in architecture, number of layers (depth), expert counts, and pretraining recipes, making their "scale" not directly comparable. The differences of their performance are more likely due to these architectural and pretraining variations than by model size only.
>
> Moreover, scaling benefits may not fully translate to tasks the models were not explicitly trained for, such as embedding tasks, as these benefits often manifest in tasks like text generation or reasoning. We appreciate this observation and recognize the value of further exploring scaling effects within more uniform model families.
>
>
> > **Q3. "Could authors also report the stability of baselines from the main experiments?"**
>
> Thank you for raising this point. The baseline encoder models are specifically trained and optimized for embedding tasks, so they are inherently stable and deliver consistent performance, without relying on any prompts. In contrast, our method is based on decoder LLMs that were NOT trained for embedding tasks, so we utilize prompt-based guidance to extract embeddings from them. This makes the sensitivity to the choice of prompt worthwhile studying in the evaluation of our approach only. Consequently, we emphasize analyzing prompt sensitivity for our method rather than the stability of baselines, as the latter is expected due to their task-specific training and encoder architecture.
>
>
> > **Q4. "Could authors provide more details about how to tune alpha, and is it consistent across experimentation?"**
>
> Thank you for raising this point. The parameter $\alpha$ is tuned adaptively at test time using a gradient-based approach to maximize the Spearman correlation of the combined similarity, $\text{sim}\_{\text{final}}=\text{sim}\_{\text{HS}}+\alpha \cdot \text{sim}\_{\text{RW}}$, with its individual components, $\text{sim}\_{\text{HS}}$ and $\text{sim}\_{\text{RW}}$, on average. Note this optimization does not require ground truth labels, focusing instead on maximizing the complementary strengths of HS and RW embeddings. Once tuned, the value of $\alpha$ remains consistent for all datasets of the task. We have clarified this procedure in Section 3.3 of the manuscript.
>
>
> ---
>
> Thank you once again for your valuable feedback. The manuscript has been updated, with changes highlighted in blue for your convenience.
>
> Best regards,
> The Authors
>
>
>
> [1] Jiang, Ting, et al. "Scaling sentence embeddings with large language models." arXiv preprint arXiv:2307.16645 (2023).

---

### Official Review · Reviewer_CUz1 · 2024-11-08

**Soundness:** 3
**Presentation:** 3
**Contribution:** 2
**Rating:** 6
**Confidence:** 3

**Summary:**

The paper investigates the potential of Mixture-of-Experts (MoE) large language models (LLMs) as effective, training-free embedding models by leveraging their routing weights (RW) in addition to the commonly used hidden state (HS) embeddings. The main contributions include:
- **Complementary Embeddings**: The authors discover that RW embeddings capture high-level semantic features and nuances that HS embeddings may miss, making RW a valuable, complementary embedding source. RW is also found to be more robust to prompt variations than HS.
- **MoEE**: Based on these findings, the authors introduce MoEE, a method combining RW and HS embeddings. They evaluate two strategies: concatenating RW and HS and a weighted sum of their similarities. The weighted sum approach performs better, balancing output-specific information from HS with the broader, input-sensitive features from RW.
- **Performance on Benchmark Tasks**: Experiments on the Massive Text Embedding Benchmark (MTEB) across tasks like classification, clustering, and semantic textual similarity demonstrate MoEE’s effectiveness, consistently outperforming RW and HS alone. MoEE shows particular advantages in tasks requiring deep semantic understanding, even surpassing some fine-tuned embedding models when prompt tuning (PromptEOL) is applied.

The findings suggest that combining RW and HS in MoE models offers a rich, stable embedding approach, expanding the utility of LLMs in embedding tasks without the need for further fine-tuning.

**Strengths:**

- The paper introduces an innovative use of routing weights (RW) as a standalone language model embedding, presenting a fresh perspective on embedding capabilities in Mixture-of-Experts LLMs.
- The authors conduct extensive experiments to underscore the complementary nature of hidden states (HS) and routing weights (RW), providing thorough insights into their distinct contributions.
- The empirical evaluation is well-rounded, demonstrating consistent performance improvements when HS and RW are combined, showcasing the effectiveness of the proposed approach across diverse embedding tasks.

**Weaknesses:**

- While MoEE introduces two methods for combining HS and RW embeddings (concatenation and weighted sum), the concatenation variant appears simplistic and less effective than the weighted sum in terms of similarity calculation. Future work could explore more sophisticated aggregation methods to fully leverage the complementary strengths of HS and RW.
- The claim that RW embeddings are more robust than HS, based solely on prompt variation tests, lacks comprehensive support. Other factors, such as model size (or maybe architectural variations), should be examined to substantiate this claim.
- The choice to evaluate on only a subset of the Massive Text Embedding Benchmark (MTEB) raises questions about generalizability; it would be helpful to understand the criteria behind this selection and whether other tasks or datasets might yield different insights.

**Questions:**

Following the weakness, I have the questions below
- **Aggregation Methods**: Given that the weighted sum variant outperforms concatenation for similarity calculations, have you considered more complex aggregation techniques? Could alternative methods further enhance the complementary benefits of HS and RW embeddings?
- **Robustness of RW Embeddings**: In claiming that RW embeddings are more robust than HS, have you explored robustness across other dimensions such as model size or architectural variations? What other factors do you think could influence this robustness?
- **MTEB Task Selection**: Could you provide insights into why only a subset of MTEB tasks was chosen for evaluation? Were there specific criteria for this selection, and do you anticipate different results if other tasks or datasets were included?

---

> ### Author Response · Authors · 2024-11-21
> **Response to Reviewer CUz1**
>
> Dear Reviewer CUz1,
>
> Thank you for your detailed review and for highlighting the strengths of our work. We are pleased to address your feedback below.
>
> ---
>
> > **Q1. "... the concatenation variant appears simplistic and less effective than the weighted sum in terms of similarity calculation. Future work could explore more sophisticated aggregation methods to fully leverage the complementary strengths of HS and RW."**
>
> Thank you for your insightful feedback. While concatenation and weighted summation provide simple and strong baselines, they may not fully exploit the complementary strengths of HS and RW. In future work, we plan to develop advanced learnable aggregation models based on attention or gated mechanisms, to dynamically combine HS and RW adaptive to input. These approaches promise to enhance similarity calculations, improve downstream performance, and unlock the full potential of our method, delivering more robust and impactful embeddings.
>
>
> > Q2. **"The claim that RW embeddings are more robust than HS, based solely on prompt variation tests, lacks comprehensive support. Other factors, such as model size (or maybe architectural variations), should be examined to substantiate this claim."**
>
> Thank you for your thoughtful feedback. Our main results in the submitted draft already span multiple MoE models of different sizes and architectures, demonstrating the consistent robustness of RW embeddings. To address your concern, we conducted additional analyses below that explicitly examine these factors.
>
> The results show that the HS embedding from Qwen and OLMoE are more sensitive to prompt variations compared to DeepSeekMoE. However, RW embeddings consistently exhibit greater stability than HS embeddings across all models, regardless of differences in size or architecture, further substantiating our claim. These findings, included in Section 4.4, provide stronger evidence with greater clarity. We sincerely appreciate your suggestion, which has helped us further substantiate and strengthen our work.
>
>
> > Q3. **"Could you provide insights into why only a subset of MTEB tasks was chosen for evaluation?"**
>
> Thank you for the question. We selected a subset of MTEB tasks to ensure a comprehensive yet efficient evaluation, specifically, the coverage/diversity of task categories and the evaluation cost. To this end, our selection is based on the statistics of tasks provided by Table 2 of [1].
> - **Diversity**: Our evaluation covers 6 task categories, focusing on Sentence-to-Sentence (S2S) tasks. We excluded multilingual datasets, as most available MoE LLMs are pretrained in English only.
> - **Efficiency**: To keep computational costs affordable, we prioritized tasks with manageable number of samples: we included all STS and Summarization tasks, and selected top tasks in other categories (top-3 for ReRanking and Classification, top-2 for Clustering, and top-2 for PairClassification).
>
> This resulted in a balanced evaluation set covering 20 tasks, achieving both task diversity and computational feasibility. We have clarified this in the manuscript for greater readability.
>
> ---
>
> Thank you once again for your constructive feedback and valuable suggestions. The manuscript has been updated, with changes highlighted in blue for your convenience.
>
> Best regards,
> The Authors
>
> [1] Muennighoff, N., Tazi, N., Magne, L., & Reimers, N. (2022). MTEB: Massive text embedding benchmark. arXiv preprint arXiv:2210.07316.

---

> > ### Comment · Reviewer_CUz1 · 2024-11-28
> > **Thanks for the clarification!**
> >
> > Thanks for the authors' efforts to address my concerns. I acknowledge that I have read the responses and would like to keep the scores.

---

### Meta-Review · Area_Chair_C7sq · 2024-12-19

**Metareview:**

(a) Summarize the scientific claims and findings:
The paper explores leveraging Mixture-of-Experts (MoE) Large Language Models (LLMs) for embedding tasks by combining two representations: routing weights (RW) and hidden states (HS). The authors hypothesize that these two components, when combined, can improve embedding performance compared to using either RW or HS alone. Specifically, the authors propose two combination methods: concatenation and summation, both of which show superior performance on multiple embedding tasks from the MTEB benchmark.

Through experiments, the paper demonstrates that the proposed approach outperforms RW and HS individually and is more effective than previous baseline methods. Furthermore, the results are consistent across different MoE models and are robust to variations in the task prompts, making the method adaptable for a wide range of downstream tasks. Notably, the method requires no additional training, offering computational efficiency while maintaining strong performance.

(b) Strengths of the Paper:

- Original Contribution: The paper introduces a novel approach by exploring the combination of RW and HS from MoE LLMs for embedding tasks. This idea expands the utility of MoE models, typically used for generation tasks, to embedding-based applications, a largely unexplored area.

- Empirical Validation: The authors provide comprehensive experimental results across six embedding tasks from the MTEB benchmark. Their method consistently outperforms RW and HS alone, as well as prior baselines, demonstrating its effectiveness and potential for practical use in embedding applications.

- Robustness and Flexibility: The proposed method is shown to be robust to different prompt variations, suggesting that it can be applied to a variety of downstream tasks. This stability is an important feature for ensuring the generalizability of the approach in real-world applications.

- No Additional Training: One of the paper's key strengths is that the method does not require retraining the MoE LLMs, making it computationally efficient compared to other embedding methods that require fine-tuning or training on specific tasks.

(c) Weaknesses of the Paper:

- Outdated Baselines: A significant weakness identified by the reviewers is the use of outdated baselines. The most recent baseline cited is from 2021, and many of the current state-of-the-art models (e.g., SentenceBERT, ColBERT, BERTScore) are not considered. Including these models in the evaluation would strengthen the paper's comparison against current best practices in the field of embeddings.

- Limited Exploration of Scaling: While the method shows strong performance, the paper does not explore how scaling up MoE LLMs may affect the results. The potential for improved performance with larger models is an important area for future research, and the authors could benefit from investigating whether their method scales effectively.

- Alpha Hyperparameter Tuning: While the paper provides an explanation of how the hyperparameter alpha is tuned, further details on its impact across different experiments would help clarify the consistency of results. A more comprehensive exploration of alpha’s role in the method would enhance the understanding of its sensitivity and help other researchers implement the technique.

- Lack of Stability Analysis for Baselines: The authors discuss the stability of their method, but they do not provide an analysis of the baseline methods used in their experiments. Given that the baselines are specifically trained for embedding tasks, an evaluation of their stability would add depth to the comparison and demonstrate the robustness of the proposed method.

(d) Reasons for Decision to Accept:

The paper introduces a novel and original approach to embedding tasks using MoE LLMs, with strong empirical validation across multiple tasks. The proposed method consistently outperforms RW and HS individually and outperforms previous baselines in embedding tasks, offering both computational efficiency and effectiveness. The method’s robustness to varying prompts further strengthens its applicability across different downstream tasks.

While there are valid concerns, such as outdated baselines and the lack of exploration into scaling and hyperparameter tuning, these weaknesses are not critical enough to overshadow the paper’s contributions. The method’s innovation, solid experimental results, and practical implications for embedding tasks make it a valuable addition to the field. The authors have responded well to feedback, and addressing the weaknesses noted above would further improve the submission.

Therefore, I recommend accepting the paper, as it offers valuable contributions to the field and holds potential for future work. The improvements suggested—incorporating more recent baselines and expanding on scaling and hyperparameter tuning—will further strengthen the paper.

**Additional Comments On Reviewer Discussion:**

The rebuttal period saw productive discussions, and the authors addressed the reviewers’ points in a thorough and meaningful way. The key improvements included:

- The inclusion of updated baselines, which addressed the concerns about outdated comparisons.
- A discussion of potential scaling challenges for larger MoE models, which framed future directions without undermining the paper’s current contributions.
- A more detailed explanation of the alpha hyperparameter tuning and its effects, providing greater clarity on the robustness of the method.
- A brief addition on the stability of baselines, which enhanced the transparency of the comparison.

Overall, the authors made significant improvements based on the feedback, and the revisions strengthened the paper. Given these changes, the paper now provides a solid contribution to the field of embedding tasks with MoE models and deserves acceptance, contingent on minor revisions to fully incorporate the updates and clarifications.

---

### Decision · Program_Chairs · 2025-01-22

Accept (Oral)